# BYTESIZED32: A Corpus and Challenge Task for Generating Task-Specific World Models Expressed as Text Games

**Ruoyao Wang**[†]**, Graham Todd**[‡]**, Xingdi Yuan**[◇]**, Ziang Xiao**[◇♠]
**Marc-Alexandre Côté**[◇]**, Peter Jansen**[†]
[†]University of Arizona    [◇]Microsoft Research Montréal
[‡]New York University    [♠]Johns Hopkins University
{ruoyaowang,pajansen}@arizona.edu
gdrtodd@nyu.edu    {eric.yuan,ziangxiao,macote}@microsoft.com

## Abstract

In this work we investigate the capacity of language models to generate explicit, interpretable, and interactive world models of scientific and common-sense reasoning tasks. We operationalize this as a task of generating text games, expressed as hundreds of lines of PYTHON code. To facilitate this task, we introduce BYTESIZED32[1], a corpus of 32 reasoning-focused text games totalling 20k lines of PYTHON code. We empirically demonstrate that GPT-4 can use these games as templates for single-shot in-context learning, successfully producing runnable games on unseen topics in 28% of cases. When allowed to self-reflect on program errors, game runnability substantially increases to 57%. While evaluating simulation fidelity is labor intensive, we introduce a suite of automated metrics to assess game fidelity, technical validity, adherence to task specifications, and winnability, showing a high-degree of agreement with expert human ratings. We pose this as a challenge task to spur further development at the juncture of world modeling and code generation.

## 1 Introduction

Simulating the world through mental models is a crucial component of human problem solving, inference, and cognition (Barsalou, 1999; Buckner and Carroll, 2007; Addis et al., 2009). Large language models (LLMs) have demonstrated precursors of this ability, such as encoding a wide range of common-sense world knowledge from their training data (Li et al., 2022a). Similarly, LLMs have been used directly as interactive world simulators in text games like AI DUNGEON (Walton, 2019), where their capacity to predict tokens in context is leveraged to convert natural language user inputs (e.g. *open treasure chest*) into plausible environmental observations (e.g. *you open the chest and find within a glittering sword*).

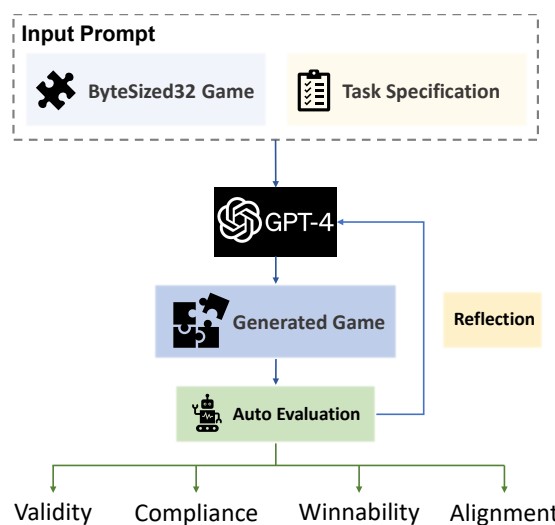

Figure 1: An overview of our text game generation and evaluation process. The model, here GPT-4, generates a game using in-context learning with a prompt consisting of (1) a single highly-templated example game, and (2) the task specification for the target game to generate. Generated games are self-reflected by providing the model with error output from a PYTHON interpreter that detects syntactic and API issues. The generated game is then evaluated by an automated evaluation harness, as well as manually by human evaluators, to measure its technical validity, specification compliance, physical reality alignment, and winnability.

In this work, we examine instead whether LLMs can be used to generate explicit and task-specific world models expressed as *code*, providing a more formal and interpretable method to examine a model's world knowledge. We operationalize this as a problem of generating the complete PYTHON source code of an interactive text game that centers around a particular common-sense task, such as *washing dishes with a dishwasher* or *building a campfire*. Although an interactive multi-step simulation of even modest tasks typically requires several hundred lines of code, we show that it is possible for LLMs to generate these simulations using single-shot in-context learning. This is accom-

---

[1]Code: github.com/cognitiveailab/BYTESIZED32

plished by providing the heavily-templated source code of an existing text game as input, and tasking models with adapting the template to a novel specification, as shown in Figure 1. The template provides a consistent, scalable, and general-purpose code architecture by hierarchically decomposing the simulation into object classes and sub-classes (e.g. device and container), which can be instantiated to make specific game objects (e.g. stove and jug). The template also offers example implementations of common actions (e.g. activating devices or opening containers) and scoring functions that automatically detect task progress.

The contributions of this work are:

1. We present BYTESIZED32, a corpus of 32 world models (expressed as text games in PYTHON) centered around tasks that require common-sense reasoning. The corpus includes 20k lines of code (including detailed comments), and is suitable for both in-context learning or producing fine-tuned models.

2. We develop a suite of metrics to assess the quality of generated games, including measuring technical aspects of the code, whether a game contains required content, how accurately a game models the physical world, and whether a game is winnable. We show that most of these metrics can be automated with a high agreement to gold human ratings, dramatically reducing the manual labor required to evaluate model-generated simulations.

3. We show that a model with a large input context, GPT-4, can produce runnable text games for unseen tasks in 28% of cases using in-context learning alone. When allowed to self-reflect on its own generated code combined with PYTHON interpreter errors that assess syntax issues or API compliance, the model dramatically increases performance, generating runnable simulations in 57% of cases.

4. We empirically demonstrate that while current best-generated games frequently include task-critical objects and actions, they only accurately model the physical world in 51% of cases, while being winnable in only 38% of cases. We pose this as a challenge task to spur further development at the juncture of world modeling and code generation.

## 2   Related Work

**Text Games and Virtual Environments:** Interactive text environments are an attractive choice

for studying embodied agents, owing to their relative simplicity compared to full 3D simulations and ability to model complex and abstract tasks (Jansen, 2021; Li et al., 2021). While early text game research focused on testing agents on a small set of extant "interactive fiction" games like *Zork*, recent approaches have leaned towards procedurally generating a wider set of simple text-based games in order to evaluate agents' ability to generalize (Côté et al., 2018; Urbanek et al., 2019; Shridhar et al., 2020; Wang et al., 2022). These frameworks typically rely on hand-crafted rules and templates programmatically arranged in novel configurations, though some efforts leverage external data sources (Barros et al., 2016) and generative language models (Fan et al., 2019) as well. In contrast, in this work we require models to produce a novel text game as a complete program, expressed as PYTHON code, using only a single existing game for reference.

**Code Generation:** As large language models have become more capable, interest in their ability to generate working snippets of program code has only grown. Several recent datasets have been proposed to facilitate this research, covering a wide range of programming languages and problem types (Yu et al., 2018; Lin et al., 2018; Austin et al., 2021; Chen et al., 2021). Contemporaneously, improvements in model architecture and training have led to impressive gains in code generation (Chen et al., 2021; Nijkamp et al., 2022; Li et al., 2022b; Fried et al., 2023). The GPT-4 language model (OpenAI, 2023), in particular, has sparked an interest in the use of prompting for code generation tasks, a technique which has led to advancements problem decomposition (Pourreza and Rafiei, 2023) and self-debugging by reflecting on errors (Chen et al., 2023; Olausson et al., 2023). Despite these gains, however, existing code generation benchmarks tend to require short and relatively simple programs. In contrast, here models must generate hundreds of lines of PYTHON code to generate complete and accurate task simulations. Similarly, we show that self-reflection can substantially increase the runnability of even large model-generated simulations.

## 3   The BYTESIZED32 Corpus

To support the task of generating simulations in the form of text games, we construct a corpus of highly-templated text games written in PYTHON

| BYTESIZED32 Corpus Statistics (per game) | |
| --- | --- |
| Lines of PYTHON code | 618.1 |
| Lines of comments | 198.1 |
| Tokens | 6792 |
| Action verbs | 9.8 |
| Valid actions | 306.6 |
| Object classes | 5.4 |
| Object instances | 7.4 |
| Expert path length | 12.8 |
| Total Games | 32 |

Table 1: Corpus statistics of BYTESIZED32. Values represent average values per game. Tokenization includes comments and was performed with `tiktoken`.

that can serve as examples in a *few-shot* generation paradigm. Each game is between 500 and 1000 lines of code (including comments), and encodes a task which requires either scientific or common-sense reasoning to complete. For example, a game might require an agent to *apply a bandage*, *load a dishwasher*, or *make a campfire*. Each game has two components: the game code itself, as well as a detailed *task specification* in the form of structured comments at the top of each game that provide a detailed summary of the game task, critical objects, actions, distractors, and a high-level solution plan. These components are described below, with corpus summary statistics provided in Table 1, and an example game playthrough shown in Figure 2.

### 3.1 Task Specification

The *task specification* is a set of structured comments at the start of each game in the corpus that serve as a high-level outline for the critical components of each game. These are intended to provide a high-level scaffold that language models can use to better structure games they generate. The components of the task specification include:

**Task Description:** The task an agent playing the game has to solve – for example, *washing dirty dishes using a dishwasher*.

**Task-Critical Objects:** Names of task-critical objects, such as *dishes*, *dish soap*, and a *dishwasher*.

**Actions:** Actions that an agent playing the game can take, such as *opening* or *closing* containers, *activating* or *deactivating* devices, *picking up* or *putting down* objects, and so forth.

**Distractors:** Objects (or actions) that limit or hinder task performance, or that are unrelated to the game task – for example, adding *food* that an agent can eat, that creates more dirty dishes.

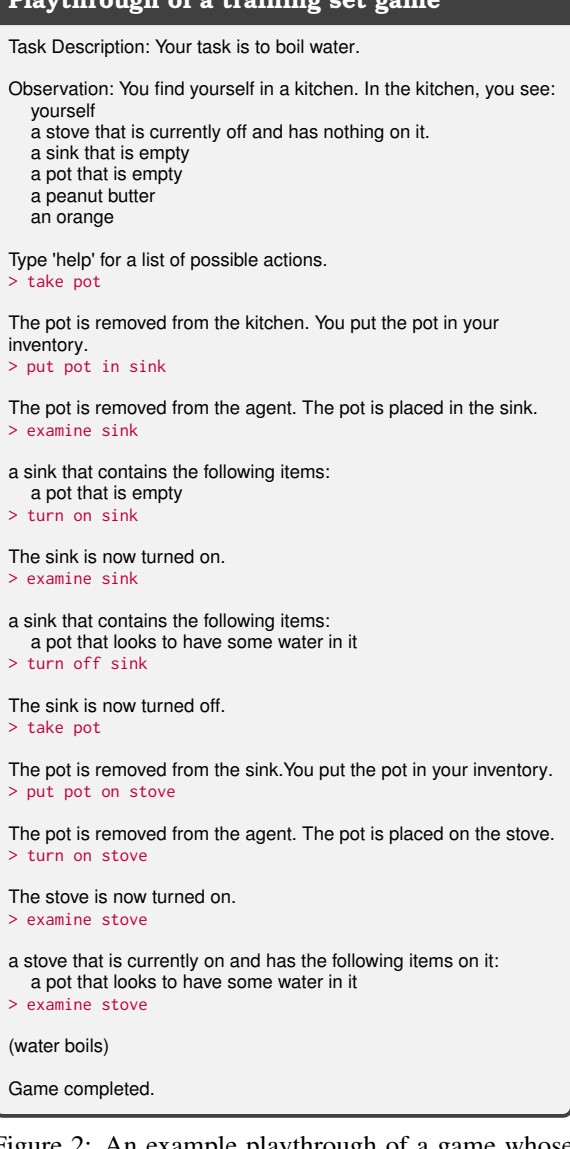

Figure 2: An example playthrough of a game whose task requires *boiling water*, from the BYTESIZED32 training set.

**Solution:** A high-level solution to the game. For example: opening the dishwasher, moving each dirty dish from the kitchen into the dishwasher, moving dish soap into the dishwasher, closing the dishwasher, and activating the dishwasher.

### 3.2 Game Code

To maximize utility as *n-shot* training data for code generation tasks, each game in the corpus uses a highly-templated structure consisting of core objects and member functions, shown in Figure 3 and described below. The core architecture and API of these functions mirrors other text game frameworks (Hausknecht et al., 2020) derived from the OPE-NAI GYM specification for reinforcement learning models (Brockman et al., 2016). These include:

**World Initialization:** Initialize the game world. For example, for the dishwasher game, create a *kitchen* room that includes *dirty dishes*, *dish soap*, a *dishwasher*, and any other relevant objects.

**Valid Actions:** Return a list of all possible valid actions that an agent could take, given the current state of the environment. For example, *take dirty dish*, or *open dishwasher*.

**Take Action Step:** Perform a specific action in the environment. This function returns the observation that results from that action – for example, the *take dirty dish* action might return the observation *"the dirty dish is now in your inventory"*.

**Scoring:** Return an agent's current progress in solving the game task, abstracted to an arbitrary numerical score, and a set of boolean flags that represent whether the game has been won or lost.

While the above methods are provided through a main game class (`TextGame`), each game also includes a large number of classes representing specific **game objects**. Each game object derives from a common class, `GameObject`, from which generic subclasses that share common methods inherit (e.g. `Containers`, that can store objects, or `Devices`, that can be activated), before finally instantiating specific game objects (e.g. `Dish`, `Dishwasher`).

## 4 Evaluating Generated Simulations

Evaluating model-generated text games presents a number of challenges. First, because games are largely open-ended and constrained only by a short task prompt, the evaluation metrics must be robust to a wide range of potential errors and behaviors. Second, evaluating open-ended simulations typically requires manual human evaluation, which is costly and labor intensive. Here, we propose a set of fully-automatic metrics that measure both technical aspects of games – such as whether the simulation runs error-free – as well as content aspects that measure how well generated simulations adhere to task specifications. We then validate these automatic instruments with human ratings.

### 4.1 Metrics

Evaluation metrics are described briefly below, with additional details of their computation, validation, and prompts described in the APPENDIX.

**Technical Validity:** The *technical validity* metric measures whether the core member functions of

**BYTESIZED32 PYTHON template**

```python
# Generic parent class for all game objects
# Provides getters/setters for object properties
class GameObject():
    ...

# Parent class for game objects that are containers
# Provides methods for adding/removing objects
class Container(GameObject):
    ...

# Parent class for game objects that are devices
# Provides methods for activating/deactivating a device
class Device(GameObject):
    ...

# Example object: Soap for washing dishes
class DishSoap(GameObject):
    ...

# Example object: A dish (that can contain food)
class Dish(Container):
    ...

# Example object: A dishwasher (that can contain dishes,
# dish soap, and be activated to wash the dishes)
class Dishwasher(Device, Container):
    ...

# Main Simulation Class
class TextGame():
    # Creates the game world and populates with game objects
    # (including the kitchen, dishes, dishwasher, etc.)
    def initializeWorld():
        ...

    # Returns a string describing the game and task
    def getTaskDescription():
        ...

    # Returns an array with all possible valid actions given
    # the current game state
    def generateValidActions():
        ...

    # Performs an action (e.g. turn on dishwasher) in the
    # environment, changing the environment state.
    def step(action:str):
        ...

    # Calculate the current game score given progress.
    def calculateScore():
        ...

# Main Entry Point (example of a user playing)
if __name__ == "__main__":
    game = TextGame()
    print("Task: " + game.getTaskDescription())
    while not game.gameOver:
        actionStr = input("> ")
        observation, score, reward = game.step(actionStr)
        print("Observation: " + observation)
        print("Score: " + score)
        print("Reward: " + reward)
    print("Game Completed.")
    print("Game Won: " + str(game.gameWon))
```

Figure 3: An illustration of the core classes and member functions present in the highly-templated games of the BYTESIZED32 corpus. Each game consists of an average of 618 lines of code, and the example here provides only an overview of a subset of the most important functions.

a generated text game run without errors by calling them in a PYTHON interpreter and capturing errors. We measure errors during the *game initialization* phase where the simulation environment is being constructed, the *valid action generation*, where the simulation provides a list of all valid actions the agent might take given the current environment state, and the *step* function, which takes a user-requested action that modifies the environ-

ment. The *valid action generation* and *step* functions are tested by exhaustively crawling all possible trajectories (i.e. sequences of actions) an agent could take. Because games can have up to 2000 valid actions per step, the path crawling procedure has a limited horizon – typically 3 steps. At each step, we also group actions by their action verb and explore a maximum of 100 actions of each group.

**Specification Compliance:** This metric measures whether the generated game includes the required actions, objects, and distractors required in the task specification. Compliance is measured automatically by supplying the generated game and its *task specification* to a GPT-4 model, which is then asked a series of true-or-false questions about the presence of each required component. For example, in a *boiling water* game, one such question is *Does the simulation contain the object 'Sink'?*. To validate this automatic metric, we compare GPT-4 ratings with gold ratings generated by two expert human annotators, showing moderate-to-strong inter-annotator agreement between GPT-4 and human ratings (Avg. $\kappa = 0.74$; Object: $\kappa = 0.96$; Action: $\kappa = 0.75$; Distractor: $\kappa = 0.50$).

**Physical Reality Alignment:** In addition to technical and specification compliance, we provide a measure of how well the actions in generated games accurately model the constraints of the physical world. For example, a game that lets you take an object out of a closed container (like a fridge) without first opening it is not respecting the constraints of the physical world. Because a simulation necessarily encodes only a small subset of reality, we restrict our measure to only the set of actions implemented by the game and returned by the GEN-ERATEPOSSIBLEACTIONS() function.

To measure physical reality alignment, we crawl a given game up to a depth of 3 steps, then randomly sample 100 trajectories equally distributed across each action a game implements. These trajectories are then provided to GPT-4, with a prompt to provide a binary assessment as to whether the game playthrough up to that point adheres to physical reality, as well as a short text justification for that assessment. To validate this metric, two expert human raters produced gold labels for 200 physical reality alignment judgements. The inter-annotator agreement between human and GPT-4 ratings is strong (Cohen's $\kappa = 0.89$), demonstrating GPT-4 has a high agreement with humans when making these assessments.

**Winnability:** A game is considered *winnable* if there exists a sequence of actions that, when performed in order, will lead to a winning state of the game. Automatic evaluation of winnability was performed by letting a GPT-4 text game agent play through the games. This agent uses recent prompting techniques such as ReAct (Yao et al., 2023) and Reflexion (Shinn et al., 2023) to provide high-level planning and problem-solving. Manual evaluation was performed by a single human evaluator. Both automatic and manual evaluators attempted to reach the game's winning state by submitting actions to the game's STEP() function. We note that this process does not always give an accurate measure of a game's winnability, as it is possible for an evaluator to fail to find a possible winning trajectory. Nevertheless, we find empirically that in the vast majority of cases a game is either obviously winnable or obviously impossible to win. Overall we find that GPT-4 underestimates game winnability, with inter-annotator agreement between GPT-4 and the human evaluator low (Cohen's $\kappa = 0.43$). This suggests solving arbitrary text games zero-shot is still beyond the capabilities of GPT-4, a finding consistent with prior research on LLMs as commonsense problem solvers (Bian et al., 2023). As such, we report human evaluations of winnability for our experiments.

## 5 Experiments

Here we investigate the capacity for a large language model such as GPT-4 [2] to generate the hundreds of lines of code required to generate a working text game simulation centered around unseen tasks on each of the technical and quality metrics described above. Alongside, we examine the extent to which these models can use reflection to increase generation performance.

### 5.1 Model and Prompt

As template games contain up to 10K tokens before including the prompt, we make use of GPT-4 (OpenAI, 2023) with a context window of 32K tokens for each of our experiments. The model prompt includes a *1-shot* example (a single PYTHON reference game from the BYTESIZED32 corpus), followed by a *task specification* describing the game the model must generate, drawn from an unseen

---

[2]See Appendix C for performance of CodeLlama (Rozière et al., 2023) on BYTESIZED32 with no finetuning.

evaluation set.[3] All experiments reported here use greedy decoding. Additional model hyperparameters and prompts can be found in the APPENDIX.

## 5.2 Evaluation Set

In addition to the 32 training games in the BYTE-SIZED32 dataset, we also provide an evaluation set in the form of *task specifications* for 16 additional unseen games. Each game in the evaluation set is explicitly crafted to have highly similar or highly dissimilar characteristics to specific games found in the training set, such as similar or dissimilar *objects*, *actions* or *distractors*.

## 5.3 Reference Game Selection

To improve diversity in generation, we randomly pair each game specification in the evaluation set with six different reference games. Half of these reference games are chosen to have at least some similarity to the evaluation game (i.e., they share a similar object, action, or distractor), while half are chosen to minimize similarity.[4] With 16 game specifications in the test set, this results in a total of 96 model-generated games.

## 5.4 Reflection

Similar to other code generation tasks (Lehman et al., 2022), we hypothesize that self-reflection – that is, providing the model with error output, and allowing it to iteratively correct its generated code – will increase overall generation performance. As such, during generation, we provide any error messages generated by the PYTHON interpreter during the *technical validity* evaluation back to the model, in a self-reflection prompt that requests the model to correct the error. This reflection step is repeated until the game passes all *technical validity* checks, or a maximum number of self-reflection steps is reached. In the experiments reported here, the maximum number of reflection steps is 3.

## 6 Results

Here, we evaluate all generated games $(N = 96)$ across each metric, reporting results both before and after self-reflection. The results of the *technical*

| Technical Validity | Number of Reflections | | | |
|---|---|---|---|---|
| Measurement | 0 | 1 | 2 | 3 |
| Game Initialization | 85.4% | 85.4% | 89.6% | 88.5% |
| Valid Actions | 80.2% | 83.3% | 87.5% | 88.5% |
| Runnable Game | 28.1% | 42.7% | 51.0% | 57.3% |

Table 2: Technical validity measurements of generated games before reflection (0), and after up to three reflection steps. Values represent the proportion of games $(N = 96)$ passing a given test after a given number of reflection steps.

| | Reflection | | |
|---|---|---|---|
| Measurement | Before | After | Δ |
| *Specification Compliance* | | | |
| Task-critical objects | 100.0% | 100.0% | 0.0% |
| Task-critical actions | 93.8% | 93.8% | 0.0% |
| Distractors | 21.9% | 18.8% | -3.1% |
| Winnability | 30.2% | 37.5% | +7.3% |

Table 3: *Specification compliance* and *winnability* measurements for generated games before and after reflection. *Specification compliance* is measured automatically, while *winnability* is measured manually by human experts. Overall, each measure shows small increases or decreases post-reflection.

*validity* evaluation are shown in Table 2. Model performance on creating game initialization methods is strong overall, beginning at 85%, and increasing to 89% after reflection. Similarly, generating a method that enumerates valid actions for a given step occurs in 80% of generated games before reflection, increasing to 89% after reflection. Generating fully runnable games, which successfully run an exhaustive path crawl of all possible game trajectories up to 3 steps without error, occurs in only 28% of games before reflection, but increases to 57% after reflection – a substantial increase of 29% gained from the self-reflection process. We show examples of GPT-4 fixing bugs in code via reflection in APPENDIX Table 7.

Self-reflection also increases *physical reality alignment*, with a histogram of automatically measured physical reality alignment scores shown in Figure 4. Before reflection, average physical reality alignment across games is 43%, which indicates that GPT-4 finds 43% of randomly sampled paths to fully comply with its expectations of physical reality. After reflection, this increases to 51%, a moderate increase of 8%. Though measured automatically, the strong inter-annotator agreement between human and GPT-4 raters in Section 4 suggests this improvement to be genuine, though it is

---

[3] We empirically observe that without providing the templated example, GPT-4 generates games that exhibit limited state space (i.e., less challenging), and lack a coherent API for the development of a consistent automatic evaluation metric. We leave *0-shot* game generation as future work.

[4] These pairings between evaluation and reference games are provided with the BYTESIZED32 corpus.

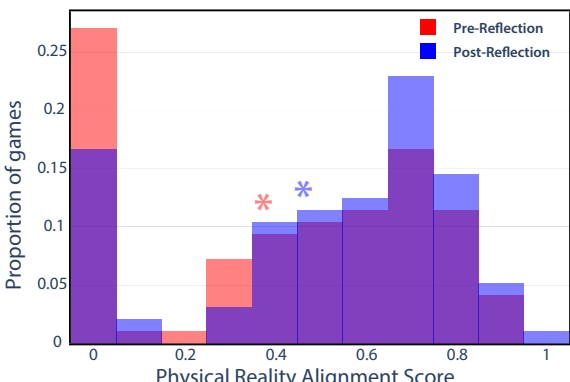

Figure 4: A histogram of automatically measured *physical reality alignment* scores, both before (red) and after (blue) reflection. Asterisks represent average scores (0.43 pre-reflection, 0.51 post-reflection).

likely due – at least in part – to an overall increase in non-zero physical reality alignment scores due to more games becoming runnable after reflection. When zero scores are removed from the analysis, the average physical reality alignment scores before and after reflection are 58% ($N = 71$) and 62% ($N = 80$), respectively – or in other words, the effect size of reflection decreases to 4%.

Similarly, self-reflection increases the *winnability of games*. As shown in Table 3, the *winnability* of all generated games pre-reflection is 30.2% when measured manually by human experts, with this increasing to 37.5% post-reflection – a gain of 7.3%.

Self-reflection does not improve every metric we evaluate. Each of the submeasures of *specification compliance*, including generating *task-critical objects*, *task-critical actions*, and *game distractors* observes a small decrease or no difference post-reflection when measured automatically – suggesting that self-reflecting on *technical validity* measures affords limited utility to these measures of simulation content. Still, we observe strong overall performance in *specification compliance*, with generated games including *task-critical objects* in nearly every case, *task-critical actions* in 93.8% of games, while struggling with distractors – generating these in only 18.8% of games.

## 7 Discussion

**To what extent can GPT-4 generate long structured text games using single-shot in-context learning?** At a high level, our results indicate that GPT-4 is frequently capable of generating syntactically valid, templated, and playable programs that

---

**Playthrough of a model-generated game**

Task Description:
Your task is to protect yourself from mosquitoes by putting on mosquito repellent, then move to the forest, take an apple, and move back to the house to put the apple in a box.

Initial Observation:
You find yourself in a house. In the house, you see:
  yourself
  a bottle of mosquito repellent
  a bottle
  a box
You also see:
  a way to the forest

Type 'help' for a list of possible actions.

> take mosquito repellent
The mosquito repellent is removed from the house. You put the mosquito repellent in your inventory.

> use mosquito repellent
You use mosquito repellent on yourself.

> move to forest
You move from house to forest.

> take apple
The apple is removed from the forest. You put the apple in your inventory.

> move to house
You move from forest to house.

> put apple in box
The apple is removed from the agent.The apple is placed in the box.

Game completed.
Game Won: True

Figure 5: An example playthrough of a GPT-4 generated game centered around the task of protecting ones self from mosquito bites. The reference game used during generation was centered on using sunscreen to protect from sunburns.

are hundreds of lines in length, such as the game shown in Figure 5. Of the generated games, nearly all implement at least one task-critical object, 88% implement at least one task-critical action, and a full 38% allow a user or agent to reach a winning state. A more nuanced interpretation of these results suggests that the model has best learned to successfully replicate the *high-level structure* of the highly-templated BYTESIZED32 game API – as model performance begins to degrade once we examine the minute details: only 58% of games are robust to a 3-step exhaustive trajectory search, and only 19% of games include a required distractor despite their presence in the reference games. Similarly, while the average training game includes 4396 code tokens, the average model-generated game contains only 3368 code tokens – or 77% of the length of training games – suggesting that model-generated games are not yet able to replicate the full level of simulation fidelity provided in the training corpus.

**How does self-reflection assist game generation?** Self-reflection – that is, iteratively providing an LLM with error messages from the PYTHON interpreter when running generated games, then asking it to correct those errors (Chen et al., 2023) – dramatically increases generation performance, most notably in terms of technical validity. We find that three steps of self-reflection increases the generation rate of runnable games from 28% to 57%. Gains from self-reflection are typically largest when detailed and specific error feedback is possible (Olausson et al., 2023). This indicates that similar gains might be achievable on errors in *specification compliance*, *physical reality alignment*, and *winnability* by using the output of our automatic evaluation process. However, the feasibility of this approach is constrained by the time and expense involved in querying the model for reflection – our results indicate that a single game could include hundreds of small and large errors in total. The cost of reflection might be lowered by the use of code *diffs* for edits (Lehman et al., 2022) or through batching multiple errors into a single reflection request – though our pilot experiments on applying these techniques to the long generated programs here indicates that current models might require specialized fine-tuning to do so. Alternatively, open source code generation models are quickly approaching GPT-4 performance in both generation length and accuracy (Li et al., 2023; Luo et al., 2023; Gunasekar et al., 2023; Rozière et al., 2023), suggesting that fine-tuning on the entire BYTESIZED32 corpus may become viable in the near-term, potentially reducing the number of errors when generating high-fidelity simulations, and reducing dependence on self-reflection.

**Can we use LLMs to automatically evaluate the output of their own simulations?** Automatic evaluation of model outputs is a vital prerequisite for large-scale experiments in world model generation, where manual evaluation of even a single simulation can require a prohibitive amount of time. The complexity of the task, however, precludes the use of most simple automatic metrics. We find that using language models to automatically crawl and evaluate generated simulations is a viable alternative to time-consuming human annotation for certain measures. This automatic evaluation is valid as long as the inter-annotator agreement between the LLM and human annotators is high. By presenting models with game code or trajectories and requesting targeted, binary judgements, we find it is possible to automatically and reliably rate measures of game *specification compliance* and *physical reality alignment*. At the same time, we show that certain metrics remain challenging for automation. Determining *winnability* of a generated game, in particular, essentially requires a model to solve arbitrary text games – an active area of research (Jansen, 2021). The automated agent currently underestimates game *winnability* by about half compared to expert human judgements, though it is plausible that this gap will narrow as the reasoning capabilities of LLMs continue to improve.

**Can we observe the internal world models of LLMs through the simulations they generate?** Generating world models as code provides a formal and interpretable means to explicitly assess how LLMs understand the world. For instance, the GPT-4 model generated a game that involved burying a box of treasure in a hole but required placing soil back into the hole *before* placing the treasure box inside. In another generated game, an agent was able to directly place water in its inventory without using any containers. These failure modes indicate situations in which the language model was unable to accurately realize the world knowledge presumably contained within its pretraining data. Even after reflection, only 51% of short 3-step trajectories in GPT-4 generated games accurately modeled the physical world, indicating that constructing correct and explicit world models in code remains a formidable challenge for LLMs.

## 8 Conclusion

In this work, we present BYTESIZED32, a corpus of small world models expressed as text games centered around specific common-sense tasks. Using a *simulation as code generation* paradigm, we show that it is possible to use these games, expressed as hundreds of lines of PYTHON code, as templates for in-context learning, and generate novel simulations for unseen tasks. We futher show that it is possible to iteratively self-reflect on these large simulations, and improve on *technical validity* and *physical reality alignment* metrics by as much as

29% and 8% respectively. While manually evaluating simulations is labor intensive, we empirically demonstrate that a number of measures of simulation accuracy can be automatically evaluated with moderate-to-strong agreement with expert human ratings. We release this work as open source, and as a challenge task at the intersection of world modeling and code generation, to spur further development in expressing the knowledge contained in language models in more formal and interpretable forms.

## Limitations

This work examines the ability of LLMs to generate abstract text-based world models and the BYTE-SIZED32 corpus is designed to facilitate that task. As such, games in the corpus are not designed to resemble extant text games or to be entertaining.

We perform our experiments in a single-shot regime and do not examine the possibility of including more than one game from the corpus could within the 32k token context window of our GPT-4 MODEL. We also do not test recent models with similar context sizes like COLT5 (Ainslie et al., 2023), UNLIMIFORMER (Bertsch et al., 2023), or CLAUDE-100K (PBC, 2023). Both of these are valuable directions for future work.

Finally, we perform reflection by regenerating the complete program at each step and target only a single error at a time. This process could be made more efficient by outputting only a code diff and batching multiple errors at once.

## Broader Impact

**Generating Simulations:** We provide an initial investigation of formalizing the knowledge captured by language models into interactive simulations expressed as code. This process can be used to inspect and evaluate language models, and the ability to generate simulations on-the-fly has potential applications in games and science.

**Self-Evaluation of GPT-4:** In spite of recent criticisms of the ability of GPT-4 to evaluate its own output, we empirically validate that this is possible in some cases where strict binary judgements of relatively straight-forward common-sense tasks are required. Automatic evaluation is a critical component of any effort at scalable environment generation, as it vastly reduces the amount of human labor required to validate and grade outputs. At the same time, strict binary measures may not be desirable for some metrics, and we leave creating and validating more granular metrics for future work.

**Self-Reflection:** Self-reflection is a rapidly emerging tool for iteratively improving the accuracy and quality of code generated by large language models. Here we show that with targeted feedback, self-reflection is possible and helpful for large (several hundred line) simulation programs. This potentially enables the generation of increasingly complex programs without sacrificing code quality.

**Intended Use:** The games included in the BYTESIZED32 corpus have been designed to study LLMs and there is no guarantee they will be entertaining or useful outside this scope.

## Acknowledgements

We thank the three anonymous reviewers for their comments. This work supported in part by National Science Foundation (NSF) award #1815948 to PJ, and the Allen Institute of Artificial Intelligence (AI2). We thank Matheus Pereira for help setting up CodeLlama-34b-Instruct and vLLM.

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

# A Details on Code Generation

In this work, we make extensive use of OpenAI's API. We use the ChatCompletion mode without system prompt. In all our experiments, we keep the following hyperparameters constant:

- temperature=0.0
- top_p=1
- frequency_penalty=0.0
- presence_penalty=0.0

## A.1 Game Generation

To generate the new games, we use the gpt-4-32k model and the following prompt.

---

**GPT-4 Game Generation Prompt**

You are DeveloperGPT, the most advanced AI developer tool on the planet. You answer any coding question, and provide real useful example code using code blocks. Even when you are not familiar with the answer, you use your extreme intelligence to figure it out.
Your task is to write a program that: is a text–based simulation. The program should be written in Python. It should be challenging to the user, testing their common–sense knowledge, and take multiple steps to complete. If possible, there should be distractor objects and actions that do not help progress, to measure whether the user really knows what they're doing. You should name all target objects and distractor objects with common–sense names. Your code must contain a class named TextGame. The TextGame class should have the following member functions:
__init__(self, randomSeed), getTaskDescription(self), generatePossibleActions(self), step(self, actionStr), calculateScore(self)

Here is a specification of the task that your code should simulate.

```# Task: Create a micro–simulation that models how to heat milk to a temperature that is suitable for a baby using a stove.
# Environment: kitchen
# Task–critical Objects: Stove, Pot, Milk, Fridge, Thermometer
# High–level object classes: Device (Stove, Fridge), Container (Stove, Pot, Fridge)
# Critical properties: temperature (Milk), temperature_increase_per_tick (Stove), temperature_decrease_per_tick (fridge), max_temperature (Stove), min_temperature (fridge)
# Actions: look, inventory, examine, take/put object, open/close container, turn on/off device, use thermometer on object, feed baby with milk
# Distractor Items: None
# Distractor Actions: drink milk
# High–level solution procedure: open fridge, take pot containing milk, put the pot on the stove, turn on the stove, use the thermometer to moniter the milk temperature till the temperature is suitable for a baby to drink, feed baby
```

Here is an example of a text–based simulation on a different topic that you can use as a template:
{**GAME_CODE**}

---

Depending on the length of prompt and the code to generate and the API traffic, each game may require 5-10 minutes to generate. We use stream generation which allows us to recover from a GPT-4 API timeout. The code in the response of GPT-4 is wrapped in a Markdown Python code block (i.e., enclosed with three backticks) which makes it easy to extract. We only keep the code part and discard the rest.

## A.2 Reflection

During validity check, when the code encounters an error from the Python interpreter, we use the error message to reflect. For the reflection we use the standard GPT-4 model (i.e., with 8k context) and the following prompt.

---

**GPT-4 Reflection Prompt**

You are DeveloperGPT, the most advanced AI developer tool on the planet. You answer any coding question, and provide real useful example code using code blocks. Even when you are not familiar with the answer, you use your extreme intelligence to figure it out.
Your task is to correct a program that is a text–based simulation. Here is the code of the simulation
```
{**GAME_CODE**}
```
Here is the error message from a Python interpreter.
{**ERROR_MESSAGE**}
You should respond all the code with your fix. Do not respond anything else.

---

# B Additional Notes on Evaluation Metrics

## B.1 Technical Validity

Validity measurements are reported in order, such that failure of a function called earlier in the API implies failure for all subsequent tests. We note, however, that the game initialization functions are evaluated only once, at the beginning of the game, while the GENERATEPOSSIBLEACTIONS() and the STEP() function are necessarily evaluated at each step.

## B.2 Specification Compliance

The full GPT-4 prompt used to generate the true-or-false evaluations of specification compliance is provided below:

---

**GPT-4 Specification Compliance Prompt**

You are DeveloperGPT, the most advanced AI developer tool on the planet. You answer any coding question, and provide real useful example code using code blocks. Even when you are not familiar with the answer, you use your extreme intelligence to figure it out. Your task is to evaluate a program that is a text–based simulation.

Here is a specification of the simulation: {**GAME_SPEC**}

Here is the code of the simulation: {**GAME_CODE**}

Answer the following question based on the given specification and the simulation code: {**EVAL_QUESTION**}

Answer 'Yes' or 'No' first and briefly explain your answer.

---

| Measurement | Manual (Human) | Automatic (GPT-4) | $\Delta$ |
|---|---|---|---|
| *Specification Compliance* | | | |
| Task-critical objects | 97.2% | 100.0% | 2.8% |
| Task-critical actions | 87.5% | 93.8% | 6.3% |
| Distractors | 37.5% | 18.8% | 18.7% |
| Winnability | 37.5% | 17.8% | 19.7% |

Table 4: Manual (human) and automatic (GPT-4) evaluation results of the best-generated games (i.e. games after reflection, $N = 96$) for both *specification compliance* and *winnability*. Difference scores reflect the difference between automatic and manual ratings, showing a moderate overall agreement for specification compliance (Avg. $\kappa = 0.74$; Object: $\kappa = 0.96$; Action: $\kappa = 0.75$; Distractor: $\kappa = 0.50$) and modest agreement for winnability ($\kappa = 0.43$).

Because we observe variance in GPT-4's answer on a few games, even when using a temperature of zero, we run the same measure 31 times and take the final result as the majority vote.

The upper part of Table 4 shows the manual and GPT-4 automatic evaluation results of specification compliance. The inter-annotator agreement between GPT-4 and the human evaluators are high (average Cohen's $\kappa = 0.71$). We note that the manual evaluation differs from the automatic evaluation in only 8 of the 96 games, indicating that automatic evaluation is a viable substitute for costly human evaluation of these metrics. We also note that the automatic evaluation most frequently differed from the manual evaluation in the distractors section, which is also the section that proved the most difficult in terms of generation.

## B.3 Physical Reality Alignment

The process of generating sample trajectories for automatic evaluations occurs in two steps. First, we perform a breadth-first crawl of the game using the action strings returned at each step by the GENER-ATEPOSSIBLEACTIONS() function. At each step we maintain a list of counts for each action "verb," which is extracted from a valid action string by taking the first token. When we perform a recursive search from a given step, we keep only 10 paths for each action verb. We restrict our search to a maximum depth of 3 actions and stop after 25,000 paths have been generated. In addition, if the game produces an error, then the error message is recorded as the observation from that step and the search continues.

After the initial set of paths has been generated, we group the set by the last action verb used in each path. We then generate a subsample of 100

paths by taking an even number of paths from each group. For instance, if the actions TAKE, PUT, and MOVE occur as the last actions in our set of 25,000 paths then we subsample 33 paths for each action and 1 path randomly. Each subsampled path is sent to GPT-4, along with the game's task description, which is then asked to determine whether every action in the path and its accompanying observation align with physical reality. The GPT-4 evaluation prompt is provided below:

---

**GPT-4 Physical Reality Alignment Prompt**

In the playthrough of the text game below, I would like you to describe whether the game engine (i.e. the observations it returns in response to actions) are physically accurate models of the world or whether they don't make sense.

An example of not making sense would be being able to take an action from a container (like a fridge) without having opened it first. In addition, if an action produces an error from the game, then it automatically fails to accurately model the world and does not make sense.

Please restrict your evaluation only to the short playthrough, and the specific actions chosen, without speculating about other actions.

Note: Objects can be manipulated by the agent without first being explicitly picked up, as long as they are in the environment, and readily accessible (e.g. not in a closed container).

The evaluation should be binary ("yes" or "no"), except in the cases where the code generated an error, when the evaluation should be "error".

Here is an example output format: {"evaluation":"no", "short_justification": "could take an object (banana) from the closed fridge without having to first open the fridge"}

Game Task: {**GAME_TASK**}

Here is the playthrough to evaluate: {**PATH**}

---

## B.4 Winnability

The initial prompt given to GPT-4 to act as a text game agent is provided in Figure 6. The GPT-4 agent was used to evaluate 96 games, and the results were broken down as follows: in 17 games, agent reached the end of the game as determined by the game's CALCULATESCORE() function. In 16 games, the GPT-4 agent finished by outputting "done," but the game's CALCULATESCORE() did not indicate that a terminal state had been reached. In 34 games, the GPT-4 agent finished by outputting "bug" (see Figure 7 for an example). In the remaining 29 games, the game crashed before the GPT-4 agent finished execution.

The bottom part of Table 4 shows the manual and GPT-4 automatic evaluation results of winnability. The inter-annotator agreement between GPT-4 and the human evaluators are low (average Cohen's $\kappa = 0.43$). As a result, we report the manual results of winnability in the main paper.

You are playing a text–based games. Interact with the environment to solve a task.

The game you are about to play only understands one command at a time from the following list of commands:
{**POSSIBLE_ACTIONS**}

Prepend your thoughts with 'think:' when planning your next steps. When you think the task is completed, say 'done'. If you think the task can't be completed at all, say 'bug'.

**The IN_CONTEXT_EXAMPLE used in the GPT-4 Winnability Prompt**

Task Description: Your task is to make ice cubes.

Initial Observation: You find yourself in a kitchen. In the kitchen, you see:
   yourself
   a freezer that is currently closed
   a sink that is currently off and that is empty
   a pot that is empty
   a peanut butter
   a orange

```
> think: I will need some container to make the ice cubes
in. I will need to fill the container with water and then
put it in the freezer. I will need to wait for the water to
freeze. I will need to remove the ice cubes from the
container.
> think: I will start by looking for a container.
> open freezer
```
The freezer is now open.

```
> examine freezer
```
a freezer that is currently open and contains the following items:
   an ice cube cray that is empty

```
> take ice cube tray
```
The ice cube tray is removed from the freezer. You put the ice cube tray in your inventory.

```
> think: I will need to fill the ice cube tray with water.
Based on the possible commands, I have to move the ice cube
tray to the sink and then turn on the sink.
> put ice cube tray in sink
```
The ice cube tray is removed from the agent.
The ice cube tray is placed in the sink.

```
> look
```
You find yourself in a kitchen. In the kitchen, you see:
   yourself
   a freezer that is currently open and empty
   a sink that is currently off and that contains one or more items.
   a pot that is empty
   a peanut butter
   a orange

```
> examine sink
```
a sink that is currently off and that contains the following items:
   an ice cube cray that is empty

```
> turn on sink
```
The sink is now turned on.

```
> examine sink
```
a sink that is currently on and that contains the following items:
   an ice cube cray that looks to have some water in it

```
> turn off sink
```
The sink is now turned off.

think: I will need to put the ice cube tray in the freezer and wait for the water to freeze.
```
> take ice cube tray
```
The ice cube tray is removed from the sink. You put the ice cube tray in your inventory.

```
> look
```
You find yourself in a kitchen. In the kitchen, you see:
   yourself
   a freezer that is currently open and empty
   a sink that is currently off and that is empty
   a pot that is empty
   a peanut butter
   a orange

```
> put ice cube tray in freezer
```
The ice cube tray is removed from the agent.
The ice cube tray is placed in the freezer.

```
> close freezer
```
The freezer is now closed.

```
> look
```
You find yourself in a kitchen. In the kitchen, you see:
   yourself
   a freezer that is currently closed
   a sink that is currently off and that is empty
   a pot that is empty
   a peanut butter
   a orange

```
> look
```
You find yourself in a kitchen. In the kitchen, you see:
   yourself
   a freezer that is currently closed
   a sink that is currently off and that is empty
   a pot that is empty
   a peanut butter
   a orange

```
> open freezer
```
The freezer is now open.

```
> examine freezer
```
a freezer that is currently open and contains the following items:
   an ice cube cray that looks to have some ice in it

Current step: 18
Score: 1
Reward: 0
Game Over: True
Game Won: True

Figure 6: Prompt used for the automatic evaluation of the winnability of generated games.

| Technical Validity | Number of Reflections | | | |
|---|---|---|---|---|
| Measurement | 0 | 1 | 2 | 3 |
| Game Initialization | 60.4 | 52.1 | 52.1 | 56.2 |
| Possible Actions Generation | 60.4 | 52.1 | 52.1 | 52.1 |
| Runnable Game | 20.8 | 20.8 | 20.8 | 20.8 |

Table 5: Technical validity measurements of games generated with CodeLlama before reflection (0), and after up to three reflection steps. Values represent the proportion of games ($N = 96$) passing a given test after a given number of reflection steps.

| Measurement | Reflection | | |
|---|---|---|---|
| | Before | After | $\Delta$ |
| *Specification Compliance* | | | |
| Task-critical objects | 96.9% | 78.1% | -18.8% |
| Task-critical actions | 100% | 90.6% | -9.4% |
| Distractors | 18.8% | 9.4% | -9.4% |
| Winnability | 4.2% | 4.2% | 0% |

Table 6: *Specification compliance* and *winnability* measurements for games generated with CodeLlama before and after reflection. *Specification compliance* is measured automatically, while *winnability* is measured manually by human experts. Overall, each measure shows non negligible decreases post-reflection due to CodeLlama's difficulty with performing meaningful reflection on large pieces of code.

# C  Open-Source Language Model

We report here the results when using the open source language model CODELLAMA-34B-INSTRUCT (Rozière et al., 2023) served using vLLM (Kwon et al., 2023) with maximum context length of 32k and a dynamic RoPE scaling factor of 2. We opted for CodeLlama because of it can extrapolate up to context size of 100k tokens. While its one-shot code generation performance is decent with 20.8% runnable games compared to 28.1% for GPT-4, this result is tempered by CodeLlama's apparent difficulty with performing meaningful reflection on these large pieces of code. As shown in Table 5, some of the revised games degrade right after the first reflection and never fully recover. The two main failure cases for doing reflection with CodeLlama is 1) partial code generation and 2) generating the full code but without including the fix even though CodeLlama's responses mention the issue and how to fix it.

# D Qualitative Examples

---

**Example of GPT-4 agent detecting an issue**

Here is the task:
You find yourself in a room. In the room, you see: . a
vending_machine. a quarter coin. a dime coin. a nickel coin. a
penny coin. yourself.
Your task is to buy a snack from the vending machine using the
coins available in the room.

> think: I need to collect the coins and use them in the
vending machine to buy a snack. The task does not specify
which snack to buy, so any snack will do.
OK.

> take quarter
The quarter is removed from the room. You put the quarter in your
inventory.

> take dime
The dime is removed from the room. You put the dime in your
inventory.

> take nickel
The nickel is removed from the room. You put the nickel in your
inventory.

> take penny
The penny is removed from the room. You put the penny in your
inventory.

> think: Now that I have all the coins, I need to use them
in the vending machine to purchase a snack.
OK.

> put quarter in vending_machine
You put the quarter in the vending machine. Total money in the
vending machine: $0.25

> put dime in vending_machine
You put the dime in the vending machine. Total money in the
vending machine: $0.35

> put nickel in vending_machine
You put the nickel in the vending machine. Total money in the
vending machine: $0.40

> put penny in vending_machine
You put the penny in the vending machine. Total money in the
vending machine: $0.41

> select chocolate
Not enough money in the vending machine to buy the chocolate.
You need $1.59 more.

> select chips
Not enough money in the vending machine to buy the chips. You
need $0.59 more.

> think: I don't have enough money to buy any snack. Task
can't be completed.
OK.

> bug

---

Figure 7: An example of using GPT-4 agent to detect
issue with a generated game.

| | |
|---|---|
| Game | 0512_action_test_9_p_gpt-4-32k_volume_generation.py |
| Error | `invalid syntax (line 293)` |

| Code diff | | | |
|---|---|---|---|
| | `293` | `defstep(self, actionStr):` | `def step(self, actionStr):` |
| | | `- - -` | |
| | `338` | `reward = self.score - lastScore` | `reward= self.score - lastScore` |

| | |
|---|---|
| Game | 0512_distractor_test_15_p_gpt-4-32k_dishwasher_generation.py |
| Error | `'TextGame' object has no attribute 'actionInventory'` |

| Code diff | | | |
|---|---|---|---|
| | `290` | | `>` `def actionInventory(self):` |
| | `291` | | `>` `inventory = self.agent.contains` |
| | `292` | | `>` `if len(inventory) == 0:` |
| | `293` | | `>` `return "You have nothing in your inventory."` |
| | `294` | | `>` `else:` |
| | `295` | | `>` `inventory_str = \` |
| | | | `"In your inventory, you have:\n"` |
| | `296` | | `>` `for item in inventory:` |
| | `297` | | `>` `inventory_str += "\t" + \` |
| | | | `item.makeDescriptionStr() + "\n"` |
| | `298` | | `>` `return inventory_str` |

| | |
|---|---|
| Game | 0512_distractor_test_16_p_gpt-4-32k_plant-tree_generation.py |
| Error | `local variable 'measuring_cup' referenced before assignment` |

| Code diff | | | |
|---|---|---|---|
| | `457` | | `>` `stone = None` |
| | `458` | | `>` `measuring_cup = None` |
| | `459` | | `>` `scale = None` |
| | | | `- - -` |
| | `468` | | `>` `if stone is not None and measuring_cup is \` |
| | | | `not None and scale is not None:` |
| | `469` | `if stone.parentContainer == measuring_cup:` | `\|` `if stone.parentContainer == measuring_cup:` |
| | `470` | `self.score += 1` | `\|` `self.score += 1` |
| | `471` | `if stone.parentContainer == scale:` | `\|` `if stone.parentContainer == scale:` |
| | `472` | `self.score += 1` | `\|` `self.score += 1` |
| | `473` | `if measuring_cup.getProperty("containsLiquid"):` | `\|` `if measuring_cup.getProperty("containsLiquid"):` |
| | `474` | `self.score += 1` | `\|` `self.score += 1` |
| | `475` | `if self.score == 3:` | `\|` `if self.score == 3:` |
| | `476` | `self.gameOver = True` | `\|` `self.gameOver = True` |
| | `477` | `self.gameWon = True` | `\|` `self.gameWon = True` |

Table 7: Examples of GPT-4 fixing bugs via reflection. Note our reflection approach generates the entire code rather than the patches, we show difference between code before and after reflection for clarity purpose.