# OpenReview forum: "ByteSized32: A Corpus and Challenge Task for Generating Task-Specific World Models Expressed as Text Games"
_EMNLP/2023/Conference — EMNLP 2023 Main_

### Official Review · Reviewer_sjFC · 2023-08-02

**Soundness:** 3

**Excitement:**

4: Strong: This paper deepens the understanding of some phenomenon or lowers the barriers to an existing research direction.

**Missing References:**

- [Implicit Representations of Meaning in Neural Language Models](https://aclanthology.org/2021.acl-long.143) (Li et al., ACL-IJCNLP 2021)
- [Language Models as Agent Models](https://aclanthology.org/2022.findings-emnlp.423) (Andreas, Findings 2022)

**Paper Topic And Main Contributions:**

This research paper investigates the language model's capability to create interactive world models for various common-sense reasoning tasks like washing dishes or starting a campfire. The authors assess this ability by generating programs that produce text-based gaming environments with the objective of completing the given task. These environments consist of devices and containers, some of which are necessary for the goal, while others serve as 'distractors' to challenge the language model. To create these games, the authors employ the GPT-4 model API using a single shot in-context learning (ICL) approach, with ICL examples representing similar games.

Moreover, the paper demonstrates that allowing GPT-4 to self-reflect on its output significantly improves the runability of the generated programs from 28% to 58% (i.e., the generation results in 0 errors). The contributions of the research are fourfold:

1. The paper presents a corpus of 32 world models expressed as text-based games implemented in Python.
2. It introduces a set of metrics to evaluate the quality of the generated games.
3. The authors emphasize the importance of self-reflection in enhancing the final quality of the games.
4. The research highlights the current limitations in generating accurate world models and creating winnable games for this particular task.

**Questions For The Authors:**

Question A. Overall did GPT-4 have higher or lower scores for Specification Compliance and Physical Reality Alignment metrics?

Question B. Why did not you experiment with any open-sourced models? Are there no alternatives to the long context input capability of GPT-4? Did you consider using methods like Longformer (Beltagy et al. 2020)? If the concern is performance, I agree that it may not be as competitive as GPT-4. However, the sole use of a closed-access API would make it very hard to replicate and improve your results.

**Reasons To Accept:**

- The idea of projecting commonsense reasoning (or world modeling) as formal language (code) generation is intuitive.
- The newly introduced corpus, BYTESIZED32, has the potential to unlock further opportunities for future research in this particular area.
- The authors support the validity of their introduced metric through assessment by expert human annotators.
- The main hypothesis of the paper is clear and well-presented, making it straightforward for readers to follow. In section 7, the authors provide explanations for the findings related to various implicit questions raised throughout the paper, such as whether LLMs can be utilized to evaluate their own simulations or if the simulations generated by LLMs accurately represent their internal worlds.

**Reasons To Reject:**

- All the experiments were conducted using GPT-4, which is understandable considering the lack of open-source models that match its performance, especially at the time of submission. However, it would be beneficial to have a few open-source models capable of handling long contexts to ensure replicability. As it stands, future researchers cannot be certain if they are building upon this work since GPT-4's performance may not be consistent.
- The above replicability concern is also true for evaluation as they use GPT4 to evaluate the quality for the following traits: Specification Compliance, Physical Reality Alignment, and Winnability.
- The authors use GPT4 to evaluate its own output. This approach raises doubts about the soundness of the evaluation. Although the authors acknowledge this issue and present inter-annotator agreements with humans in the Discussion section, I am still uneasy about this circular evaluation. I agree with the authors' desire to automate the evaluation for large-scale experiments, but I believe the generation and evaluation should be assigned to separate models. It seems counterintuitive for a model to critique its own output, especially after multiple rounds of reflection.

**Reproducibility:**

3: Could reproduce the results with some difficulty. The settings of parameters are underspecified or subjectively determined; the training/evaluation data are not widely available.

**Reviewer Confidence:**

4: Quite sure. I tried to check the important points carefully. It's unlikely, though conceivable, that I missed something that should affect my ratings.

**Typos Grammar Style And Presentation Improvements:**

-

---

> ### Author Rebuttal · Authors · 2023-08-29
>
> **Reviewer: “All the experiments were conducted using GPT-4, [...] it would be beneficial to have a few open-source models capable of handling long contexts to ensure replicability.”**
>
> **Response:** We agree whole-heartedly with the importance of and desire for performant open-source models for this task. However, as you point out, there is simply at present a large gap in terms of both performance and capacity between closed-source and open-source large language models. Nevertheless, in order to address these concerns we have performed some preliminary investigations on recently-released open-source models. We began by investigating StarCoder, but were limited by its context size of 8k tokens. We attempted to compress our game prompts within this limit, but the resulting prompts failed to produce any outputs that complied with our specifications, making further automatic evaluation impossible. On August 24th, Meta released the new set of CodeLlama models. We used the CodeLlama-34b-Instruct model, slightly modified our prompts in order to comply with its expected format, and computed our automatic technical validity metrics. These preliminary results are presented below, and we intend to run the rest of the evaluation suite for the camera-ready submission.
>
> —-------------------------------------------------------------------
> | Technical Validity Measurement |  gpt-4 | CodeLlama-34b-Instruct |
> | -------------------------------|--------|------------------------|
> | Game Initialization            | 0.854  |          0.573         |
> | Valid Actions  	               | 0.802  |          0.552         |
> | Runnable Game                  | 0.281  |          0.198         |
> —-------------------------------------------------------------------
>
> We see that while this model is capable of producing technically valid games, it continues to lag GPT-4 in terms of performance. It’s possible that further tuning and further models will continue to narrow this gap, and we eagerly anticipate such improvements as future work. With regards to Longformer, while it’s possible that it could handle the long context lengths required by the ByteSized32 corpus, we point out that our experiments are all performed in-context without any fine-tuning. It seems unlikely that a model pre-trained in 2020 has the capacity to perform few-shot text game generation.
>
> As for the potential volatility of GPT-4’s performance: while there is some possibility that model updates could lead to a change in its ability to complete this task, we feel that the main contribution of our paper is not the particular results of GPT-4, but the introduction of a new corpus, challenge task, and evaluation suite. Especially as open-source models improve in quality, the potential impact of this volatility will decrease.
>
> **Reviewer: “The above replicability concern is also true for evaluation as they use GPT4 to evaluate the quality for the following traits: Specification Compliance, Physical Reality Alignment, and Winnability.”**
>
> **Response:** The volatility of GPT-4 is potentially more of a concern for evaluation, as future work completed at different times might be evaluated with different checkpoints. However, we (a) believe that as future model checkpoints become more performant, evaluation performance will only increase, and (b) believe that any concerns regarding automatic evaluation performance can be ameliorated by continued periodic validation with human annotators. The relative cost of this human validation compared to the time saved by automatic evaluation means that this is not a prohibitive burden on future work.
>
> **Reviewer: “Overall did GPT-4 have higher or lower scores for Specification Compliance and Physical Reality Alignment metrics?”**
>
> **Response:** We would like to ask the reviewer to elaborate on this question so we can better understand and address it.
>
> **Reviewer: “The authors use GPT4 to evaluate its own output. This approach raises doubts about the soundness of the evaluation. [...] I agree with the authors' desire to automate the evaluation for large-scale experiments, but I believe the generation and evaluation should be assigned to separate models. It seems counterintuitive for a model to critique its own output, especially after multiple rounds of reflection.”**
>
> **Response:** While we understand and sympathize with the concerns of having a model both generate and critique its own output, we feel that the particular nature of our evaluations serves to alleviate this issue. Specifically, we designed each of our evaluations to be as narrow as possible: instead of holistically evaluating the entire game, we largely focus on discrete, binary questions which drastically limit the possibility for a model to “bias” its ratings. And, perhaps more importantly, we validate the performance of these evaluations by measuring their fidelity to expert human raters, which is the gold standard in validating research instruments. In the case where the model evaluations diverged from these ratings (winnability), we did not use them. We feel that this is the gold standard for the validation of a new instrument of any type. Finally, the process of self-reflection (which is inherently a form of model self-evaluation) has been demonstrated to improve downstream performance on a variety of tasks (Reflexion: Language Agents with Verbal Reinforcement Learning, Shinn et al., 2023) – and indeed we show self- reflection to improve the performance of this task by as much as 30% (Table 2), which is a strong validation of self-reflection. (As a related note, humans also use self-reflection to great effect when learning; see e.g. the work of cognitive scientist Michelene Chi)
>
> **Reviewer: “Soundness Score of 2, Replicability Score of 2”**
> **Response:** The reject criteria for this review centers around using closed (GPT-4) vs open source models.  While we’re very sympathetic to developing and using open source models, as we have shown, that is not easily possible for this task because it’s on the very edge of the  capabilities for even the most performant LLM that exists (that just happens to be closed source).  But, in the spirit of open science, during the author response period we have worked to train a state-of-the-art code open code generation model (CodeLlama) on this task, and preliminary experiments show it strongly underperforms GPT-4.  We believe that our methods, dataset, and evaluation are model agnostic, and have made every effort to create high-quality, easily-replicable open source work that will be made freely available on Github upon acceptance.  We look forward to a day in the near-term (perhaps even within the next 6 months) where open source models can use this data to become as performant as GPT-4 is now.  In light of our going the extra mile to address all your concerns regarding closed versus open models, we ask that you consider changing your Soundness and Reproducibility scores from a hard reject (2) to something you feel more appropriate.

---

### Official Review · Reviewer_QfRB · 2023-08-05

**Typos Grammar Style And Presentation Improvements:** Overall very good writing! Simple to …
**Soundness:** 4

**Excitement:**

4: Strong: This paper deepens the understanding of some phenomenon or lowers the barriers to an existing research direction.

**Missing References:**

Metafor: https://alumni.media.mit.edu/~hugo/publications/drafts/IUI2005-metafor.4.pdf
For some of the reasoning capabilities, it would be good to maybe cite this: https://arxiv.org/abs/2303.16421 as it shows that chatgpt has some reasoning of commonsense actions, but is (as you found in your work) very limited.


**Paper Topic And Main Contributions:**

The authors provide a very interesting dataset and benchmarks for representing commonsense simple situations as executable text-based games. The authors evaluate the usage of LLMs for creating/solving/fixing these kinds of text based games. The games generated from language models trained on the dataset provide some insight as to how LLMs represent the world from text, and can help to understand how to improve reasoning in these models.

**Questions For The Authors:**

A) What would be some approaches to diminishing the amount of tokens in this work?
B) What issues do you foresee happening when people try to build on this work

**Reasons To Accept:**

* The dataset is novel and the simplicity and understandability of what is being done is very clear
* The authors give some valid automated evaluation metrics for future development
* The work, although simple, can be utilized as a very good starting point for future development. For example, what if we supply some extra world knowledge/knowledge graphs into these games? I believe that the research that can stem from this is very good


**Reasons To Reject:**

* The authors focus solely on utilizing GPT-4 which can be limiting for future development of the work
* Additionally, the context windows that this dataset needs are exceptionally high, over 4k on average according to line 470, which once more can be very limiting for future development


**Reproducibility:**

5: Could easily reproduce the results.

**Reviewer Confidence:**

4: Quite sure. I tried to check the important points carefully. It's unlikely, though conceivable, that I missed something that should affect my ratings.

---

> ### Author Rebuttal · Authors · 2023-08-29
>
> **Reviewer: “Focusing solely on utilizing GPT-4 which can be limiting for future development of the work.”**
>
> **Response:** At the time of submission, GPT-4 was the only model that could generate meaningful programs simply based on a high-level game specification. We are optimistic about the future development of this work as this area is rapidly developing. We are excited to see the release of powerful open-source models.
>
> On August 24th, Meta released the new set of CodeLlama models which claims to have a 100k context window. We used the CodeLlama-34b-Instruct model, slightly modified our prompts in order to comply with its expected format, and computed our automatic technical validity metrics. The results are as follows:
>
> —-------------------------------------------------------------------
> | Technical Validity Measurement |  gpt-4 | CodeLlama-34b-Instruct |
> | -------------------------------|--------|------------------------|
> | Game Initialization            | 0.854  |          0.573         |
> | Valid Actions  	               | 0.802  |          0.552         |
> | Runnable Game                  | 0.281  |          0.198         |
> —-------------------------------------------------------------------
>
> While there is still a large gap between CodeLlama and GPT-4, CodeLlama is able to generate non-trivial results and generate some valid games. Our hope is that our world-modeling-as-simulation task and data are ready for when each class of model -- open source or closed source -- is able to best make use of this work.
>
>
> **Reviewer: “[...] context windows that this dataset needs are exceptionally high, over 4k on average according to line 470 [...]”**
>
> **Response:** While it is true that this task requires very long (8k+) context windows, we see this as aspirational work at the edge of current model capabilities, which move very quickly.  While the best open source code generation models had ~4k context windows when we submitted this work, already this week we're starting to see models (e.g. CodaLlama) that claim to have 100k context windows.
>
> **Reviewer: “What would be some approaches to diminishing the amount of tokens in this work?”**
>
> **Response:** This is a good question.  We have experimented with removing comments (which are approximately half of all tokens), though this impacts the spirit of this work, which is to provide well-architected and highly commented code templates for world modeling in a form that models can best use to learn to structure and implement their own simulations.  Ultimately, while some hard-coding could certainly improve the efficiency of our templates and generations, they would likely be much less generalizable.  With the current transition to large (e.g. 100k+) context models, we don't view the current context length as a limitation, and are excited to explore what even larger (and higher-fidelity) world model generations are possible in the future.

---

### Official Review · Reviewer_nYh3 · 2023-08-05

**Typos Grammar Style And Presentation Improvements:** 1.	Line 137
**Soundness:** 4

**Excitement:**

4: Strong: This paper deepens the understanding of some phenomenon or lowers the barriers to an existing research direction.

**Paper Topic And Main Contributions:**

This paper evaluates the capacity of LMs to generate world models expressed as Python code. A highly templated one-shot example is provided in context and the generated game is evaluated on many dimensions. Further, reflection (re-prompting with compiler feedback) is used to iteratively change generated code.

**Questions For The Authors:**

A.	(See 1 above) Analysis of performance between similar vs dissimilar reference games? Analysis of the diversity of different codes present in the dataset to show the requirement of code generation beyond simple template matching. How do the codes in the dataset differ wrt programming concepts, semantics, etc?

B.	The scoring function seems to be the most complex and open-ended code to be generated. What is the performance analysis on the scoring function generated? How is a valid scoring function defined, etc?

C.	(See 2 above) Sensitivity to prompt design? Explicitly asking to include distractors in the prompt?

D.	(See 3 above) Error analysis of code generated.

**Reasons To Accept:**

1.	Relevant and interesting problem setup evaluating the reasoning and world knowledge of LLMs through code generation of text-based games.
2.	Code generated is hundreds of lines long, much longer than usual code generation problems.
3.	Promising results on preliminary evaluation of LM capabilities to generate game code, relevant direction of future research.

**Reasons To Reject:**

1.	Diversity in code generation: Since the dataset code is highly templated, it is unclear if the LM is template matching to generate the test game or has to perform non-trivial reasoning and code generation.
a.	Authors should provide an analysis of differences in code in the test set and one-shot example used. Line 368 mentions a setup where the reference game provided is dissimilar to the target game. However, no results are shown on LLM performance when similar vs dissimilar reference games are provided.
2.	Since the input is highly templated, an analysis could be provided on model sensitivity to prompt design. For example, one issue highlighted is the low performance of LLMs to include distractors in the generated code. The prompt shown in Appendix A.1 has the words, “If possible, there should be distractor objects…”. Does changing the prompt language to explicitly require distractors help?
3.	Error analysis is lacking: What are the common issues in generated code, parallels which can be drawn to reasoning capabilities, etc?

**Reproducibility:**

4: Could mostly reproduce the results, but there may be some variation because of sample variance or minor variations in their interpretation of the protocol or method.

**Reviewer Confidence:**

3: Pretty sure, but there's a chance I missed something. Although I have a good feel for this area in general, I did not carefully check the paper's details, e.g., the math, experimental design, or novelty.

---

> ### Author Rebuttal · Authors · 2023-08-29
>
> **Reviewer: “Diversity in code generation: Since the dataset code is highly templated, it is unclear if the LM is template matching to generate the test game or has to perform non-trivial reasoning and code generation. [...] Line 368 mentions a setup where the reference game provided is dissimilar to the target game. However, no results are shown on LLM performance when similar vs dissimilar reference games are provided.”**
>
> **Response:** Empirically, we find that GPT-4 fails to produce long form meaningful text game code without the aid of a robust template. In the zero-shot setting, GPT-4 tends to hard-code game states without actually setting up a modular environment, a capability it appears to readily demonstrate in the few-shot setting.
>
> To your second point about the similar and dissimilar reference games: when we say that two games are “similar,” we mean that the task specifications of the  two games match either in terms of task-critical objects, task-critical actions, or both. Such an alignment between a reference task specification and the desired output task specification might help the model to some extent, but it is not possible for the model to trivially copy-paste from the one-shot example to complete the task. New objects, actions, and scoring functions need to be defined, and even objects existing in the one-shot example need to be modified when necessary.
> Our pilot experiments showed that the model compliance performance of the “similar” setting and “dissimilar” setting was not significantly different, which underscores that the model is not relying on highly similar reference templates to generate the bulk of its performance.  This underscores our overall hypothesis, that the model appears to learn the general structure of the task from the template (such as how to architect a world model as a text game in general, including how to describe, populate, and inherit game objects represented as classes), rather than copying-and-pasting specific components of the reference game into the generated game.
>
> **Reviewer: “...analysis could be provided on model sensitivity to prompt design…”**
>
> **Response:** This is an interesting point, and it is true that the choice of prompts can exert a large influence over the performance of LLMs like GPT-4. However, unlike hyperparameters, it’s difficult to perform a thorough (or even non ad-hoc) search over the space of possible prompts. We found some improvements in our prompt selection over the course of the work, but there indeed remains the possibility that a different prompt would lead to even higher performance. However, we feel that the primary contribution of this paper is not about the specific implementational details to optimize the performance of GPT-4, but rather to present a new task on world modeling (that requires generating very large, ~400 line simulations in Python), paired with a suite of validated instruments and metrics for automatically evaluating the performance of those simulations.  Further prompt improvements (and improvements to overall task performance) are welcome in future work, but we feel that they’re beyond the scope of this paper.
>
> **Reviewer: “Error analysis is lacking: What are the common issues in generated code, parallels which can be drawn to reasoning capabilities, etc?”**
>
> **Response:** Our detailed evaluation metrics serve as an error analysis by nature (and, moreover, the size of the action space in each game necessitates using automated evaluations, since manual error analyses would be either strongly limited in scope or intractable). The four metrics we introduce in the paper are the four aspects that we believe are most important to a correct text game. The validity checks the syntactic correctness, the compliance checks whether task requirements are met, the winnability checks whether the game goal is correctly set, and the alignment checks whether the game correctly simulates the physical world. Model weakness can be reflected by a low score on any of these metrics. Specifically, regarding “the common issues in generated code”, GPT-4 gets a low alignment score due to its weakness at adding physical constraints to objects (e.g. some generated games allow the agent to take objects out of a closed container). Such a weakness is a reflection of the model’s common sense reasoning capability. More discussion of this can be found towards the end of the Discussion section in the paper.
>
> **Reviewer: “The scoring function seems to be the most complex and open-ended code to be generated. What is the performance analysis on the scoring function generated? How is a valid scoring function defined, etc?”**
>
> **Response:** This is a great question – the scoring function is indeed one of the more interesting and challenging game components to generate and evaluate. Nevertheless, we evaluate it in a few ways. First, our validity check determines whether the scoring function is syntactically correct and callable. Second, our winnability check gives a sense of whether the scoring function is feasible, since (for human evaluation) we require that it return a completed score when the human expert believes the task is completed, and (for automatic evaluation) we require that it returns a completed score when our ReAct/Reflexion-based planning agent believes the game should be completed. Finally, the physical reality alignment check will flag situations where an agent was given a positive score in a way that violates our simulation. While this is far from an exhaustive check of the scoring function’s soundness, we believe that it is a strong and sufficient contribution to the state-of-the-art..
>
> In practice (as we describe in the paper): (a) For nearly every generated game, the games were either obviously winnable or obviously unwinnable to the expert human evaluator, and (b) our automated metrics tend to underestimate winnability compared to expert human judgements (k=0.43), so we report a full human evaluation of winnability so that performance isn’t underestimated in this category.  We suspect that as agents become better common-sense reasoners, and better able to automatically play the generated games, their capacity for automatically evaluating scoring functions and winnability will increase proportionally.

---

### Meta-Review · Area_Chair_suqL · 2023-09-19

**Recommendation:** 4

**Metareview:**

The authors provide an interesting dataset and benchmarks for representing commonsense simple situations as executable text-based games.
The 2 out of 3 reviewers selected "Strong" for soundness score (1 selected "Good")
Reproducibility has also enough score

---

### Decision · Program_Chairs · 2023-10-07

**Decision:**

Accept-Main

**Comment:**

The authors provide an interesting dataset and benchmarks for representing commonsense simple situations as executable text-based games.
The 2 out of 3 reviewers selected "Strong" for soundness score (1 selected "Good")
Reproducibility has also enough score